# Tools of *Aggregatibacter actinomycetemcomitans* to Evade the Host Response

**DOI:** 10.3390/jcm8071079

**Published:** 2019-07-22

**Authors:** Jan Oscarsson, Rolf Claesson, Mark Lindholm, Carola Höglund Åberg, Anders Johansson

**Affiliations:** 1Department of Odontology, Oral Microbiology, Umeå University, S-90187 Umeå, Sweden; 2Department of Odontology, Molecular Periodontology, Umeå University, S-901 87 Umeå, Sweden

**Keywords:** *Aggregatibacter actinomycetemcomitans*, invasiveness, leukotoxin, cytolethal distending toxin, serum resistance, outer membrane vesicles

## Abstract

Periodontitis is an infection-induced inflammatory disease that affects the tooth supporting tissues, i.e., bone and connective tissues. The initiation and progression of this disease depend on dysbiotic ecological changes in the oral microbiome, thereby affecting the severity of disease through multiple immune-inflammatory responses. *Aggregatibacter actinomycetemcomitans* is a facultative anaerobic Gram-negative bacterium associated with such cellular and molecular mechanisms associated with the pathogenesis of periodontitis. In the present review, we outline virulence mechanisms that help the bacterium to escape the host response. These properties include invasiveness, secretion of exotoxins, serum resistance, and release of outer membrane vesicles. Virulence properties of *A. actinomycetemcomitans* that can contribute to treatment resistance in the infected individuals and upon translocation to the circulation, also induce pathogenic mechanisms associated with several systemic diseases.

## 1. Introduction

*Aggregatibacter actinomycetemcomitans* is an opportunistic pathogen associated with aggressive forms of periodontitis that affect young individuals [1,2]. The bacterium colonizes the oral mucosa early in life and is inherited by vertical transmission from close relatives [3]. Colonization of *A. actinomycetemcomitans* on the mucosa is not associated with disease but is considered as a risk factor for translocation of the organism to the gingival margin [4]. Bacteria that colonize this ecological niche have the potential to initiate periodontal diseases if they are allowed to stay, proliferate, and express virulence factors [5,6]. *A. actinomycetemcomitans* is a facultative anaerobic Gram-negative bacterium with the capacity to produce a number of virulence factors, and it exhibits a large genetic diversity [2,7]. This bacterium is an early colonizer in the disease process, and resists oxygen and hydrogen peroxide, but is later often replaced by more strict anaerobes in the deep periodontal pocket [1]. In addition to colonizing the oral cavity, systemic translocation of this bacterium is frequently reported [8,9]. *A. actinomycetemcomitans* expresses adhesins that allow colonization of to the tooth surface and the oral epithelium, as well as to mature supragingival plaque [2]. The bacterium is described as an organism that utilizes the other inhabitants in the biofilm for its survival and utilizes metabolic products from other inhabitants of the biofilm for survival and growth [1]. In addition, it is suggested that *A. actinomycetemcomitans* can promote the overgrowth of other bacterial species, which can result in local host dysbiosis and susceptibility to infection [1]. The association of *A. actinomycetemcomitans* to systemic diseases includes endocarditis, cardiovascular diseases, diabetes, Alzheimer´s disease, and rheumatoid arthritis [10,11,12,13,14,15]. The mechanisms behind these associations are not known, but several virulence properties of *A. actinomycetemcomitans*, such as tissue invasiveness, exotoxin production, serum resistance, and outer membrane vesicle secretion are potential weapons [16,17,18,19]. In the sections below, we will further address and discuss the tools behind the ability of this bacterium to evade and suppress the host immune response. The aim of the present review is to identify and describe virulence mechanisms of *A. actinomycetemcomitans*, which are associated with immune subversion, as well as bacterial pathogenicity.

## 2. Invasive Properties

Invasion of periodontal tissues by different bacterial species has been reported in human periodontitis for several decades [20,21,22]. The study by Saglie and co-workers [22] observed the prevalence and gingival localization of *A. actinomycetemcomitans* in periodontal lesions of young patients. Transmission electron microscopic examination showed microcolonies of small Gram-negative rods in the connective tissue, as well as single bacterial cells between collagen fibers and in areas of cell debris [20]. In addition to these intra-tissue bacterial cells, bacteria were also found within phagocytic cells, which had invaded the gingival connective tissue. More recent studies have demonstrated invasion of *A. actinomycetemcomitans* into epithelial cells in vitro [23,24,25]. Interestingly, some *A. actinomycetemcomitans* genotypes have been suggested to have different tissue invasive-properties [23]. If this difference in invasive properties interferes with the ability of *A. actinomycetemcomitans* to cause various periodontal or systemic diseases is not known.

A number of *A. actinomycetemcomitans* factors that likely contribute to host cell invasion have been elucidated. These include the *tad* (tight adherence) gene locus, which mediates adhesion and is required for virulence in a rat model for periodontal disease [26]. OmpA1 (also known as Omp29) is associated with the entry of *A. actinomycetemcomitans* into gingival epithelial cells by up-regulating F-actin rearrangement via the FAK signaling pathway [27], and Omp100 (also known as ApiA) promotes adhesion of *A. actinomycetemcomitans* cells, and their invasion of human gingival keratinocytes [28,29]. It has also been described that bacteria that express a cytolethal distending toxin, such as *A. actinomycetemcomitans,* can cause disruption of the epithelial barrier and promote tissue invasion [30]. A role of *A. actinomycetemcomitans* invasion in immune modulation is supported by in vitro evidence of a subsequent induction of pro-inflammatory cytokine production, and/or apoptosis, in epithelial cells and macrophage-like cells, respectively [31,32].

Severe extra-oral infections caused by *A. actinomycetemcomitans* include brain abscesses, meningitis, septicemia, urinary tract infections, osteomyelitis, and endocarditis [18,33,34,35]. Whether the systemic translocation through the epithelial barrier is due to an active invasive process, or a result of a passive leakage into the blood stream is not known [36,37]. Bacterial invasion of the periodontal tissues has been suggested as a relevant stage in the etiopathogenesis of periodontal disease, however, there is insufficient evidence to support or exclude this mechanism as a key step in periodontal disease [38,39]. Despite the lack of conclusive studies, the invasive properties of *A. actinomycetemcomitans* have the potential to help the bacterium to evade mechanical and chemical strategies immune responses, and mechanical or chemical eradication strategies [40] (Figure 1). If these properties of the bacterium contribute to systemic translocation and survival has not yet been studied. Evidently, major reasons for bacteremia caused by oral bacteria such as *A. actinomycetemcomitans* are gingival inflammation and mechanical manipulation [41].

## 3. Production of Exotoxins

*A. actinomycetemcomitans* is the only bacterium colonizing the oral cavity known to produce two exotoxins [42,43], leukotoxin (LtxA) that specifically induces killing of human leukocytes, and a cytolethal distending toxin (CDT), which is a genotoxin, causing growth arrest by affecting DNA in proliferating cells [16,44]. Both toxins are highly conserved and occur also in several other Gram-negative pathogens [45,46]. As a sign of the large genetic diversity of *A. actinomycetemcomitans*, there are strains, representing various genotypes, which produce highly different levels of these toxins [2,6,47]. All hitherto studied *A. actinomycetemcomitans* strains carry a complete *ltxCABD* gene locus, encoding for LtxA, activation and secretion [6]. Mutations, i.e., deletions and insertions in the *ltx* promoter region have been shown to influence leukotoxin production in *A. actinomycetemcomitans* [48,49,50]. The so called JP2 genotype, which harbors a 530-bp deletion in the *ltx* promoter region is well studied, and known to be strongly associated with disease risk in the individuals carrying it [5,6,51]. Its high LtxA production is considered to be an important factor for the enhanced pathogenicity of this genotype [52].

LtxA induces several pathogenic mechanisms in human leukocytes that can all be linked to the progression of periodontal disease [16]. A substantial humoral immune response against LtxA is initiated in all the infected individuals [53,54]. LtxA kills immune cells and protects the bacterium from phagocytic killing [55]. Neutrophils exposed to LtxA activate degranulation, concomitant with an extracellular release of proteolytic enzymes and metalloproteases, such as elastase and metallproteases [56,57]. LtxA can also affect human macrophages by activating the inflammasome complex, which results in the activation and secretion of pro-inflammatory enzymes (i.e., IL-1β and IL-18) [58]. In this context, an interesting observation was recently made that *A. actinomycetemcomitans* expresses an outer membrane lipoprotein, which binds host cytokines, including IL-1β [59]. The IL-1β binding protein was designated bacterial interleukin receptor I (BilRI) and has the ability to internalize IL-1β into the viable bacterial biofilm [60]. Taken together, the abilities of LtxA to cause a proteolytic environment that can degrade immunoproteins, internalize inflammatory proteins, and kill immune cells may all contribute to the survival of *A. actinomycetemcomitans* in the infected host.

The CDT is expressed by a majority of *A. actinomycetemcomitans* genotypes, even though some of them lack a complete gene operon for expression of an active holo-toxin [45]. In vitro and in vivo studies have shown that CDT affects cellular physiology involved in inflammation, immune response modulation, and causes tissue damage [61,62]. The holo-toxin consists of three subunits (CdtA, B, and C) and is transported to the nucleus of the mammalian target cells [63]. Cells exposed for CDT induce a growth arrest followed by apoptotic cell death [44,63]. In cultures of periodontal fibroblasts, CDT induces expression of cytokines and the osteoclast activating protein, receptor activator of nuclear factor kappa-Β ligand (RANKL) [64,65]. The role of CDT in the pathogenesis of periodontitis is not entirely clear, and the literature contains studies that demonstrate an association, as well as those reporting no correlation [42,66,67].

Together LtxA and CDT can act as strong weapons against the immune response raised against *A. actinomycetemcomitans*. They can cause an imbalance in the host response by activating inflammation, killing immune cells, affecting antigen presenting cells, and inhibiting lymphocyte proliferation (Figure 2). The impact of these exotoxins in the pathogenesis of periodontal disease is apparent, whereas their role in systemic diseases is not known. However, LtxA-exposed neutrophils do release net-like structures and express patterns of citrullinated proteins that are similar to those observed in synovial fluid from inflamed joints [12,68]. Moreover, antibiotic treatment of a periodontitis patient, infected with the highly leukotoxic JP2 genotype of *A. actinomycetemcomitans*, and suffering from rheumatoid arthritis, was also cured of the joint pain after the treatment [69]. These observations indicate that *A. actinomycetemcomitans* is an interesting organism in the etiopathogenesis of rheumatoid arthritis.

## 4. Serum Resistance

Serum resistance represents an important virulence factor of bacteria that enter into the bloodstream and cause infection, allowing the bacterial cells to evade the innate immune defense mechanisms present in serum, including the complement system and antimicrobial peptides [70,71,72]. The recognition of bacterial products mediating serum resistance, therefore, represents an approach to the vaccine and drug development [73,74]. Resistance to complement-mediated killing by human serum appears to be important for *A. actinomycetemcomitans* virulence, and is a common characteristic among strains of this species, although they typically do not form capsules [28,75]. The outer membrane protein, Omp100 (ApiA), was earlier demonstrated to be important for serum resistance in some serotype b and d strains and to physically interact with and trap the alternative complement pathway negative regulator, Factor H, in vitro [28,76] (Figure 3).

Evidently, Omps produced by *A. actinomycetemcomitans* strains are immuno-reactive in the human host [77]. As the presence of antibodies towards bacterial antigens such as Omps is a known trigger of classical complement activation [78], serum resistance of *A. actinomycetemcomitans* strains would be expected to also include mechanisms interacting with this activation. We recently presented evidence that the major outer membrane protein, OmpA1, is critical for serum survival in the *A. actinomycetemcomitans* serotype a model strain, D7SS [17]. Outer membrane integrity may be one mechanism behind OmpA-mediated serum resistance in Gram-negative bacteria [79]. Interestingly, serum resistant *ompA1* mutants were fortuitously obtained, which expressed increased levels of the paralogue, OmpA2. Thus, OmpA2 can apparently operate as a functional homologue to OmpA1 in *A. actinomycetemcomitans*, and both proteins seemingly act, at least partly by binding and trapping of C4-binding protein [17] (Figure 3), which is an inhibitor of classical and mannose-binding lectin (MBL) complement activation [80]. Further to these activation pathways, alternative complement activation is needed to fully eliminate serum-sensitive *ompA* mutant *A. actinomycetemcomitans* derivatives [17]. It is plausible that serum resistance in this species, similar to in *Acinetobacter baumanii* [81], is highly complex and relies on a large number of gene products, including host factors. For example, whether cleavage of the complement molecule C3 by elastase [82], which release is triggered by leukotoxin [57], may contribute to *A. actinomycetemcomitans* serum resistance is not known. Moreover, albeit *A. actinomycetemcomitans* strains are ubiquitously serum resistant, strains not expressing their immunodominant, serotype-specific polysaccharide (S-PA) antigen are occasionally isolated [83]. As speculated previously [83], the lack of S-PA expression may represent a mechanism to evade from antibody-based host responses, which could be advantageous in blood circulation. However, an inconsistency with this notion is that the absence of S-PA expression in *A. actinomycetemcomitans* appears to be scarce.

## 5. Outer Membrane Vesicles

Outer Membrane Vesicles (OMVs) of Gram-negative bacteria are spherical membrane-enclosed nanostructures that are released from the outer membrane. They can operate as a fundamental mechanism for discharging proteins and additional bacterial components into the surrounding environment and to target host cells [84,85]. Evidence from in vitro experiments shows that *A. actinomycetemcomitans* OMVs can deliver an abundance of biologically active virulence factors to host cells, and which can modulate the immune response (Figure 4).

One such example is CDT, which is delivered into HeLa cells and human gingival fibroblasts via OMVs [86]. OMVs are also involved in the export of LtxA, peptidoglycan-associated lipoprotein (Pal), and the chaperonin GroEL to host cells [87,88,89,90]. Proteomics and Western blot analysis of *A. actinomycetemcomitans* OMVs has identified additional proteins that can contribute to evasion of the immune defense, including the IL1β-binding lipoprotein, BilRI, the outer membrane proteins Omp100, OmpA1, and OmpA2, and a Factor H-binding protein homologue [17,91,92]. A functional role in the interaction with complement by vesicles is supported by observations that *A. actinomycetemcomitans* OMVs in an OmpA1-dependent manner can bind to the classical and MBL complement inhibitor, C4-binding protein [17]. It has also been demonstrated that *A. actinomycetemcomitans* OMVs can carry small molecules, including lipopolysaccharide (LPS), which can interact with complement [93]. LPS may also play a role in the observed binding of *A. actinomycetemcomitans* OMVs to IL-8 [94]. Evidence that *A. actinomycetemcomitans* OMVs carry NOD1- and NOD2-active peptidoglycan, which can be internalized into non-phagocytic human cells including gingival fibroblasts [19], reveals a role of the vesicles as a trigger of innate immunity. Moreover, OMV-dependent release of microRNA-size small RNAs (msRNAs), may potentially represent a mechanism to transfer a novel class of bacterial signaling molecules into host cells [95]. It is not completely understood how *A. actinomycetemcomitans* OMVs may physically interact with and/or enter into human host cells to enhance bacterial evasion of the immune defense. The OMVs appear to enter into human cells via clathrin-mediated endocytosis [19,96], but can also fuse with host membranes in a process dependent on cholesterol [86]. Toxins exported via OMVs can function as adhesins in receptor-mediated endocytosis of the vesicles [97], albeit neither CDT nor leukotoxin are required *per se* for the OMV uptake into host cells [86,87]. Concomitantly, although LtxA has an apparent localization on the *A. actinomycetemcomitans* OMV surface, its receptor LFA-1 is not required for delivering the toxin into host cells [98].

## 6. Conclusions

We have summarized current knowledge regarding major attributes and strategies of *A. actinomycetemcomitans*, allowing this organism to evade the host response. Without doubt, the numerous virulence properties of *A. actinomycetemcomitans* can be linked to the pathogenesis of periodontal disease [99]. Utilization of these properties for systemic translocation of *A. actinomycetemcomitans* and its subsequent survival in this new environment has been excellently summarized and illustrated [1,100]. It is today hypothesized that the virulence characteristics of *A. actinomycetemcomitans* allow this organism to induce an immune subversion that tip the balance from homeostasis over to disease in oral and/or extra-oral sites [101]. Hence, in order to prohibit the negative systemic consequences that are associated with periodontitis, successful treatment in an early phase of the disease is fundamental. Development of specific diagnostic tools for assessment of periodontal pathogens and inflammatory components in the saliva of young individuals might make it possible to prevent the disease before its onset.

## Figures and Tables

**Figure 1 jcm-08-01079-f001:**
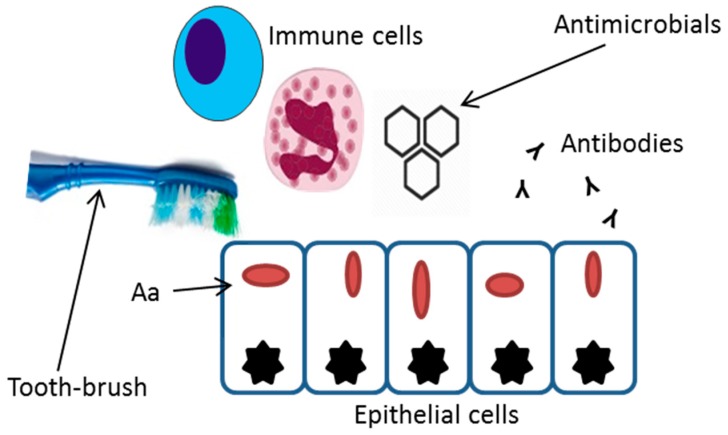
Invasion of *A. actinomycetemcomitans* into epithelial cells can protect the bacterial cells from mechanical removal, antibiotics, immune cell phagocytosis, and antibody binding.

**Figure 2 jcm-08-01079-f002:**
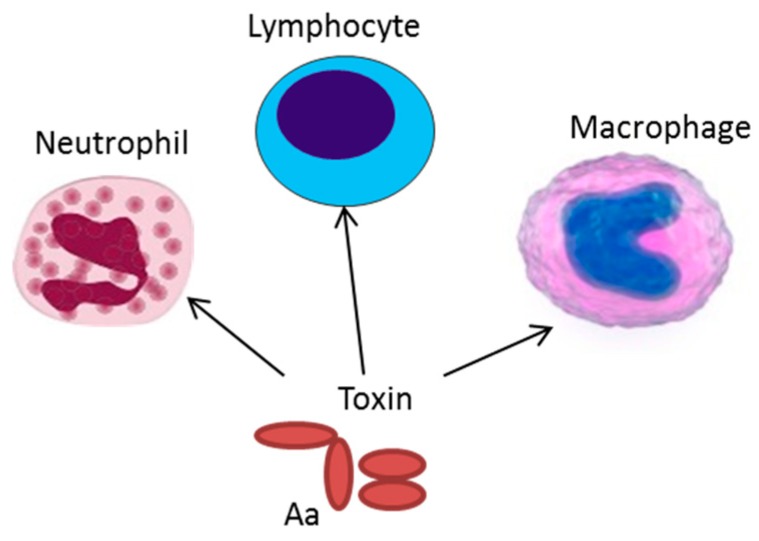
Expression of exotoxins can result in resistance of *A. actinomycetemcomitans* to phagocytosis and neutrophil degranulation. The cytolethal distending toxin (CDT) can cause inhibited proliferation of stimulated lymphocytes, and LtxA induces an inflammatory cell death in the antigen presenting cells, macrophages, and monocytes.

**Figure 3 jcm-08-01079-f003:**
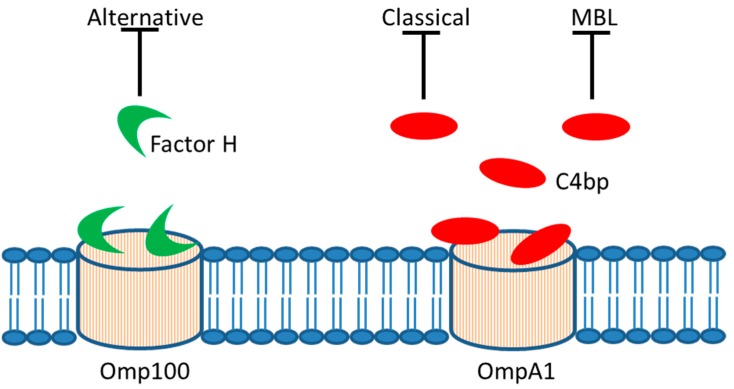
Serum resistance. In vitro evidence supports that *A. actinomycetemcomitans* outer membrane proteins, Omp100 and OmpA1, may allow trapping of soluble repressors of complement activation, i.e., Factor H and C4 binding protein (C4bp), respectively. This would result in downregulation of the alternative, classical, and mannose-binding lectin (MBL) pathway of complement activation.

**Figure 4 jcm-08-01079-f004:**
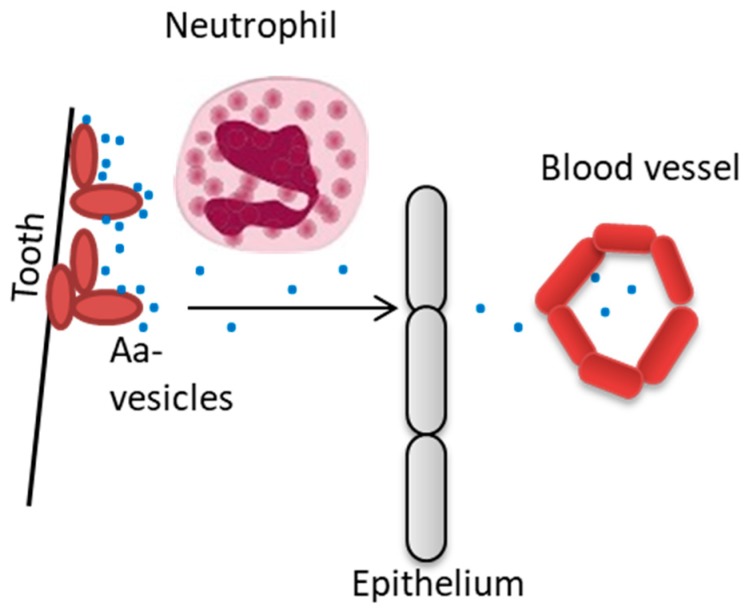
Release of outer membrane vesicles may serve as protection of *A. actinomycetemcomitans* from phagocytic- and serum killing, and also as a means to transport virulence factors to tissues that not are in close contact with the infecting bacteria.

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
