# Peer review of "Tools of Aggregatibacter actinomycetemcomitans to Evade the Host Response"

_jcm, 2019, doi:10.3390/jcm8071079_

Reviewer 1 Report

The manuscript is a well-written submission about contribution of virulence properties of A.a in periodontal disease.

 Comments and suggestions

 -             As the authors stated in line 35, 36 in page1, this bacteria is an early colonizer in the disease process. Description of the role of A.a to mature dental plaque can reinforce this fact. In addition, the corporation with other periodontal pathogen is necessary function to be an early colonizer. Please add these points to the manuscript.

Author Response

Dear reviewer (1),

Thank you for your constructive criticism on our review and for giving us the opportunity to respond to the suggestions. Below we have responded to all of the statement and suggestions point by point.

Author response to reviewer 1

1.       Aa – an early colonizer in the disease process. Description of the role of Aa to mature dental plaque?

These are two very relevant suggestions and the following information has been added to the revised manuscript, line 38. “A. actinomycetemcomitans expresses adhesins that allow colonization to the tooth surface and the oral epithelium, as well as to mature supragingival plaque [2]. The bacterium is described as an organism that utilizes the other inhabitants in the biofilm for its survival through and utilizes metabolic products from other inhabitants of the biofilm for survival and growth [1]. In addition, it is suggested that A. actinomycetemcomitans can promote overgrowth of other bacterial species which can result in local host dysbiosis and susceptibility to infection [1].”

Reviewer 2 Report

The authors present a narrative review on the tools that make A. Actinomycetemcomitans (A.A.) virulent. The authors do not mention that objective/purpose of the review in the introduction. Furthermore, the authors fail to provide fundamental information onmaterials/methods, what are the questions they are trying to answer. Even if narrative and not systematic, the authors should have provided the following information:

The authors should describe their search strategy including MeSH terms, number of studies identified, and PICO question if such where used. The authors fail to provide the following information in methodology – 
How many papers were searched -number?
How many excluded and how many included
Good quality studies/strong evidence studies included in the review

Furthermore, the authors only list the tools of A.A without mentioning how relevant that, in details, to periodontal disease and/or systematic disease (no example of relevant systematic diseases or the type of periodontal disease has been even mentioned). This makes the implications of A.A. tools unclear. 

Regarding the figures, it is not clear whether the authors have developed the figures themselves or they were taken from other scientific sources. 

Based on the previous reasons, the reviewer unfortunatelycannot recommend this review paper for publication. 

Author Response

Dear reviewer (2),

Thank you for your constructive criticism on our review and for giving us the opportunity to respond to the suggestions. Below we have responded to all of the statement and suggestions point by point.

Author response to reviewer 2

1.       The authors do not mention that objective/purpose of the review in the introduction.

In order to address this criticism we have added the following sentence to the revised manuscript.

Line 49. “The aim of the present review is to identify and describe virulence mechanisms of A. actinomycetemcomitans, which are associated to immune subversion, as well as bacterial pathogenicity.”

2.       The reviewer asks for fundamental information concerning our strategy for literature search. Since this is a narrative review, it is not common that such information is provided. In the present paper we have summarized virulence properties of Aa that contributes to its ability to evade host response. We have looked through previous review papers published in this journal, as well as other scientific journals, and we can only find such information in the systematic reviews. Therefore, we pending on editors advice before adding this information to the revised manuscript.

3.       How relevant are the properties of Aa to evade host response for its pathogenicity for different periodontal and systemic diseases.

We thank the reviewer for this very relevant question that is central for the present manuscript. However, the focus of the present review is on cellular and molecular virulence mechanisms that are associated to the etiopathogenesis of different periodontal and systemic diseases. Even though we have already discussed topics addressing the role of Aa in the pathogenesis of various periodontal and systemic diseases in the manuscript (line 40-43, 61, 74-77, 97-99, 101-102, 134-135, 179-181, 223-225, 227-229), we have added the following information to the revised manuscript, in order to further clarify these topics. Line 62 “If this difference in invasive properties interferes with the ability of A. actinomycetemcomitans to cause various periodontal or systemic diseases is not known”. Line 141 “These observations indicate that A. actinomycetemcomitans is an interesting organism in the etiopathogenesis of rheumatoid arthritis.”

 4.       Concerning sources for the figures, they are all developed specifically for the present paper. We therefore have no sources that we can refer to.

Round  2

Reviewer 2 Report

The authors have addressed the comments raised in the previous round of revision. However, this reviewer finds it unacceptable to justify not adding information on the search startegy method that this is the journal style. Indeed this is a narrative review, however, it should follow certain search strategy to make sure that this review is comprehensive and unbiased. The authors claim that this is also the case with narrative review from other journals. However, the reviewer disagrees; as an example, take a look on narrative reviewes from the Journal of Clinical Periodontology, the latest one being published on June, 2019. Therefore, the reviewer disagrees that information on search methods is only provided in systematic reviews. Therefore, the reviewer cannot recommend this review for publication, as it has no information on the search strategy, which is fundamental, even in narrative reviews.